# AMPipe: Accelerating MoE Model Training with Intra-Block Pipelining

## Abstract

The Mixture-of-Experts (MoE) architecture presents a compelling adaptation for expanding the model size of pre-trained models, such as large language models (LLMs), to enhance overall model performance (e.g., lower perplexity). However, as the sequence length (represented as $N$) increases, both the execution time of the attention layer ($O(N^2)$) and the all-to-all communication time of the MoE layer ($O(N)$, with a significant coefficient) become training bottlenecks. Current training systems have primarily focused on either optimizing the MoE layer (e.g., Tutel) or enhancing the attention layer (e.g., FlashAttention), yet they have demonstrated bounded performance improvements when confronted with long sequences—an essential consideration for modeling a potent language model with a long context window.

In this paper, we introduce AMPipe, a novel pipeline system and paragdim for accelerating the training of large MoE models using Intra-Block Pipelining, particularly when dealing with lengthy sequences. AMPipe smartly optimizes two bottlenecks together by dividing and pipelining both the attention layer and MoE layer to strategically mitigate the time costs associated with these operations. Experimental results illustrate that AMPipe can consistently outperform current training systems, which solely focus on optimizing either the MoE or attention layer. Notably, AMPipe enhances the training throughput of a highly optimized transformer block by an average of 23% across 56 benchmark cases and by up to 41% in long sequence training, all without introducing statistical impact on model convergence. Our code is available at https://github.com/iclr24-3434/AMPipe.git

## 1 Introduction

LLMs have garnered widespread attention globally nowadays (Ouyang et al., 2022; Bai et al., 2022; Touvron et al., 2023a;b; OpenAI, 2023). This surge in popularity can be attributed to their remarkable natural language generation capabilities. These LLMs are all rooted in the transformer architecture (Vaswani et al., 2017) and are constructed using stacked **Transformer Blocks**. However, it is worth noting that harnessing their advanced language abilities necessitates an exceptionally substantial investment in computational resources. Training these LLMs entails extensive utilization of powerful GPU clusters over prolonged durations. For example, the LLaMA model underwent training on a staggering 2048 GPUs for about 21 days (Touvron et al., 2023a).

In the pursuit of mitigating the substantial training costs associated with scaling up model size, the Mixture-of-Experts (MoE) paradigm (Shazeer et al., 2017; Fedus et al., 2022) has proven instrumental. This approach enables the scaling of model size without an equivalent increase in computational expenses by efficiently routing tokens to target experts. Within the general MoE-based transformer framework, the MLP layer is transformed into an MoE layer (§ 2.2). When it comes to distributed training with an expert parallelism paradigm, this MoE layer encompasses a gate function responsible for token routing, all-to-all communication mechanisms to facilitate token exchange between devices (e.g., GPUs), experts (e.g., MLPs) dedicated to token-wise computations, and a subsequent all-to-all step to return tokens to their originating devices.

The MoE architecture allows for more substantial model capacity through an increase in the number of parameters, all without imposing additional computational burdens. Consequently, this paradigm has garnered widespread adoption within giant technology companies for the training of larger lan-

guage models within reasonable budgetary constraints, ultimately leading to enhanced model quality (Fedus et al., 2022; Zoph et al., 2022; Du et al., 2022; Artetxe et al., 2021; Costa-jussà et al., 2022; Shen et al., 2023). Even GPT-4 (OpenAI, 2023) is unofficially reported to use MoE paradigm.

In the quest for improved language understanding and generation, modern LLMs have set their sights on longer context windows, particularly when dealing with extensive documents and narratives. This endeavor has driven a substantial increase in sequence length, exemplified by models such as GPT-4 (OpenAI, 2023) with 32k tokens, MPT with 65k tokens, and Claude (Bai et al., 2022) extending to a remarkable 100k tokens. However, it becomes evident that this augmentation in sequence length poses significant challenges for MoE-based LLMs. Notably, we observe that as the sequence length ($N$) increases, both the all-to-all communication and attention layers within the transformer block emerge as critical bottlenecks in the training process (as illustrated in Figure 1, left). Specifically, the time complexity of the attention layer scales quadratically at $O(N^2)$, while the all-to-all communication exhibits a linear growth at $O(N)$ but with a considerable coefficient due to the substantial volume of communication required and the relatively low communication bandwidth.

Current solutions have typically addressed these two bottlenecks in isolation, focusing on optimizing either the attention layer or the all-to-all communication.

The first category focuses on **optimizing the attention layer**. Several variants (Child et al., 2019; Beltagy et al., 2020; Wang et al., 2020; Kitaev et al., 2020) of the attention (§2.1) have been proposed to reduce the computational cost of attention layers in transformer. However, despite their high training throughput, these variations of attention implementation exhibit lower model performance. To address this issue, FlashAttention (Dao et al., 2022; Dao, 2023) has been introduced with IO-aware fusion optimization. It significantly improves the slow training speed of attention without altering convergence and effectively reduces memory usage to a linear level. Nonetheless, FlashAttention still demonstrates a superlinear time-increasing trend as sequence length grows, which is an unavoidable limitation.

We refer to the second category of solutions as **Intra-MoE Pipeline**, which serves as an MoE optimization technique aimed at resolving the all-to-all communication problem. When the all-to-all communication cost is inevitable, these costs are overlapped with the MLP computation cost in a pipeline fashion. All-to-all communication and MLP computation are divided into smaller shards and pipelined to decrease the overall cost (He et al., 2022; Hwang et al., 2023; Zhang et al., 2023). For instance, the pipeline implementation in Tutel (Hwang et al. (2023)) demonstrated significant speedup compared to its naive counterpart in vision model training tasks. However, despite achieving perfect overlap in training on Microsoft's high-bandwidth GPU clusters (1,600 Gbps), our observations indicate that these Intra-MoE Pipelines do not account for the fact that all-to-all communication is greatly slower than MLP computation, particularly on commercial clouds with a 400 Gbps network. This time gap becomes even more pronounced as sequence length increases (see Figure 1, right).

These two categories *solely* optimize one bottleneck in MoE-based transformer model training. Simply putting them together still ignores the potential of optimizing these two bottlenecks *together*. They ignore the fact that both the executions of the attention layer and MoE layers are **chunkable** along the sequence dimension. Aiming at lowering the overall cost of the attention and MoE layer, we propose an **Intra-Block pipeline** system and paragdim AMPIPE, standing for **A**ttention-**M**oE **Pipe**line. Our core idea is to chunk the attention and MoE layers into smaller shards and pipeline them to overlap the computation and communication cost inside a transformer block instead of only in an MoE layer like Tutel. We then further evaluate and analyze the costs and the speedups. This design will not change the convergence of the original model but can largely save training time and training costs.

Our contributions are stated as follows:

- We have identified and analyzed the two primary bottlenecks encountered while training large MoE-based LLMs with long sequences: (1) the attention layer with quadratic time complexity and (2) the MoE layer with linear time complexity but a significant coefficient. Our further analysis shows that the current solutions solely optimize two bottlenecks, ignoring optimization chances.

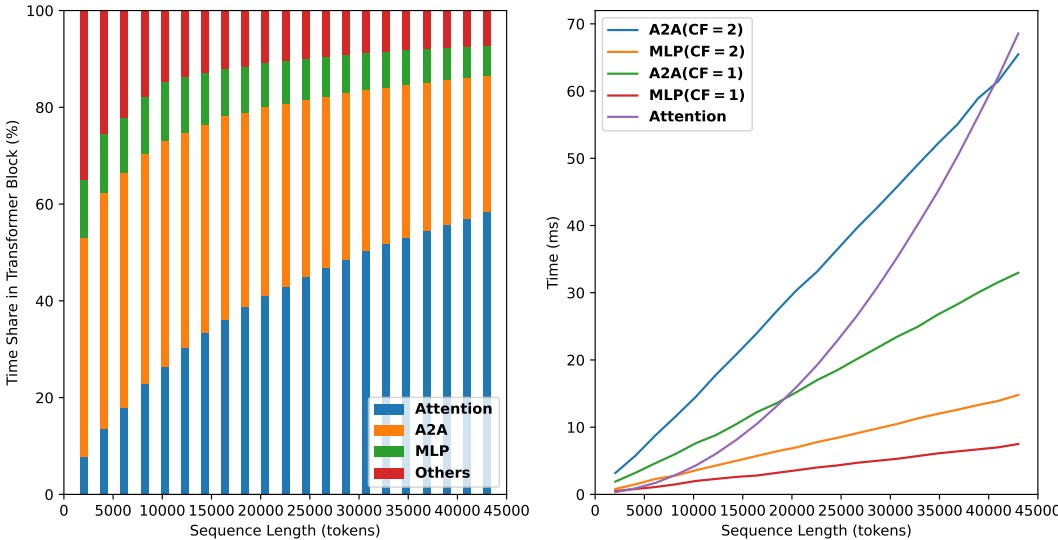

Figure 1: The execution times on our testbed for A2A (i.e., all-to-all communication), MLP, and Attention (using FlashAttention) are measured, with varying capacity factors (CF in the right subfigure) to assess the time implications of different communication volumes and computational demands.

- To address these two bottlenecks in one shot, we propose our simple yet effective solution AMPIPE. Notably, AMPIPE has been designed to cater to the needs of nearly all real-world MoE-based transformer training scenarios, making it compatible with a wide range of MoE-based language models and vision models. It stands ready for deployment, offering cost-effective and time-saving benefits, particularly for large MoE model training on **commodity cloud platforms** (e.g., with 20GB/s communication bandwidth).

- With an implementation consisting of more than 1k lines of code (LoCs), we conducted extensive evaluations within the context of real-world large MoE model training, specifically with GPT-MoE using the Megatron-LM codebase. Our evaluation results consistently demonstrate that AMPIPE surpasses current optimization techniques, achieving an average throughput improvement of 23% and a maximum throughput boost of 41%.

We provide background information in §2. We introduce the key implementation of AMPIPE in §3. Evaluation results are presented in §4. Limitations and potential future applications of AMPIPE are discussed in §A. We review related works in §B. Finally, we conclude this paper in §5.

## 2 BACKGROUND AND MOTIVATION

### 2.1 TRANSFORMER AND ATTENTION

Originally introduced in 2017 for machine translation tasks, transformer (Vaswani et al., 2017) models comprise Encoder and Decoder layers. Each of these layers incorporates **attention** and **MLP** sublayers. Attention is responsible for calculating interactions between queries and key-value pairs. Given input embeddings $\mathbf{Q}, \mathbf{K}, \mathbf{V} \in \mathbb{R}^{N \times d}$, where $N$ represents the sequence length and $d$ is the head dimension, the attention computes $\mathbf{S} = \mathbf{Q}\mathbf{K}^T \in \mathbb{R}^{N \times N}$, applies a softmax function to obtain $\mathbf{P} = \mathbf{softmax}(\mathbf{S}) \in \mathbb{R}^{N \times N}$, and outputs $\mathbf{O} = \mathbf{PV} \in \mathbb{R}^{N \times d}$. The quadratic complexity related to $\mathbf{S}$ is a bottleneck. However, modern GPUs can effectively compute attention in parallel, mitigating this bottleneck and allowing transformer models to achieve remarkable performance gains when scaled up.

Contemporary LLMs are often Decoder-only architectures (Brown et al., 2020; Zhang et al., 2022; Touvron et al., 2023a;b), primarily due to their proven language modeling capabilities when scaling up (Kaplan et al., 2020). The scalability and parallelism of attention are critical, as models exceed

billions of parameters despite their quadratic time cost. The Transformer architecture, though elegantly simple, has demonstrated its immense power, particularly when coupled with increases in model and dataset size.

## 2.2 MIXTURE OF EXPERTS

Mixture of Experts (MoE) is a model structure initially introduced in the 1990s (Jacobs et al., 1991). It regained prominence in 2017 (Shazeer et al., 2017) when applied to LSTM (Hochreiter & Schmidhuber, 1997) and more recently in 2022 (Fedus et al., 2022) for training LLMs. The fundamental concept behind MoE involves dynamically routing different portions of input data to separate expert neural networks (often MLPs).

For each input token $x$, the gate computes its probability $p_i(x)$ of routing to the $i$-th expert. It's important to note that not all tokens are selected by every expert. When applying a Top-1 gate, each token is exclusively assigned to one expert. The output $y$ of the MoE layer for input token $x$ is a weighted mixture of expert outputs, with weights determined by these probabilities:

$$y = \sum_{i \in \tau} p_i(x) E_i(x)$$

Here, $E_i(x)$ represents the output of expert $i$ for input $x$, and $\tau$ is the set of experts chosen to process this input token.

To meet system-level constraints (e.g., preventing out-of-memory errors and maintaining balanced routing), the concept of expert capacity is often applied (Fedus et al., 2022; Mustafa et al., 2022). Each expert is limited to processing a specified number of tokens, and any tokens exceeding this limit are discarded. In the Tutel design (Hwang et al., 2023), a capacity factor is introduced to efficiently adjust the expert capacities. In this approach, all experts' token inputs are padded to capacity, ensuring uniform and efficient communication and computation.

In MoE-based transformer models, the traditional MLPs are replaced with dedicated MoE layers (see Figure 2). This adaptation allows for training larger models by distributing computation across experts. The gating mechanism efficiently routes tokens to experts specializing in distinct patterns, such as different parts of speech (Zoph et al., 2022)). MoE exhibits great promise for scaling up model sizes and performance.

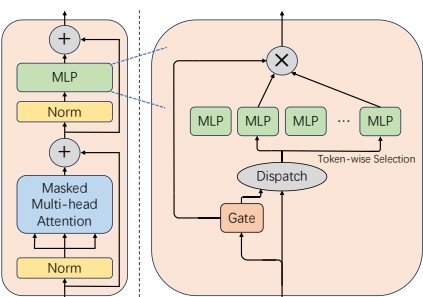

Figure 2: Transformer Block and MoE Layer

## 2.3 DATA, TENSOR, PIPELINE AND EXPERT PARALLELISM

LLMs with billions to trillions of parameters present challenges for training on a single GPU due to their immense computational and memory requirements. To address this, various parallelism techniques have been devised to distribute computation and memory effectively:

- Data Parallelism (DP): This approach shards the training data across multiple GPUs, and gradients are aggregated through an all-reduce operation during the backward pass (Li et al., 2020; Xing et al., 2015).
- Tensor Parallelism (TP): In TP, a single transformer block is divided across multiple GPUs to reduce memory usage and computation per GPU (Shoeybi et al., 2019).

- Pipeline Parallelism (PP): PP distributes different transformer blocks along model depth across different GPUs, creating an asynchronous pipeline (Huang et al., 2019; Narayanan et al., 2019; Li et al., 2021; Zhao et al., 2022).
- Expert Parallelism (EP): In MoE model training, EP is crucial, where experts within the same MoE layer are placed on the same pipeline stage but distributed across data parallel workers Fedus et al. (2022); Lepikhin et al. (2020).

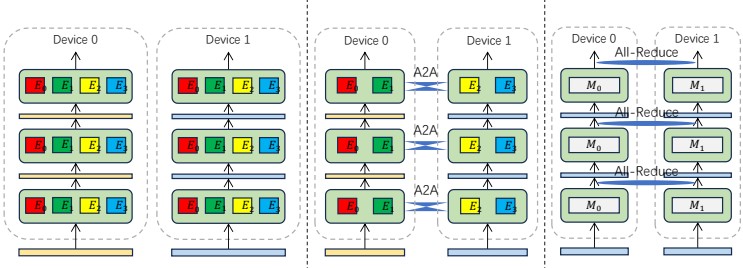

Figure 3: Data, Expert and Tensor Parallelism.

Figure 3 visually illustrates the sharding of the model across different devices using DP (left), EP (middle), and TP (right). While DP replicates the entire model on different devices, EP and TP require model data splitting across devices. DP and EP often share the same dimension. DP incurs no additional cost in the forward pass, whereas EP and TP introduce all-to-all and all-reduce communications, respectively. EP still involves extra all-to-all communication in the backward pass, while DP and TP incur all-reduce costs.

PP, on the other hand, shards the model data along the model depth, orthogonal to DP, EP, and TP. As shown in Figure 4, previous research has carefully considered pipeline bubbles and designed various PP schedules along the pipeline and time dimensions (Huang et al., 2019; Narayanan et al., 2021; Li et al., 2023a). Our work, AMPIPE, primarily focuses on accelerating the transformer block within the DP, EP, and TP dimensions and is orthogonal to these pipeline designs.

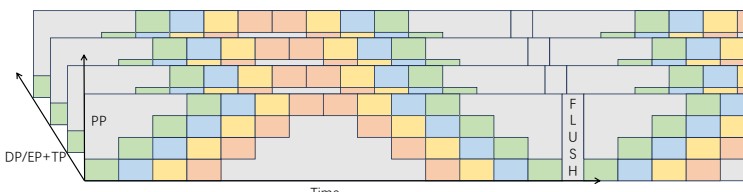

Figure 4: Inter-block Pipeline Parallelism

The state-of-the-art combination of DP, TP, EP and PP allows for the distribution of massive MoE models across numerous GPUs, making their training feasible and efficient (§B).

## 3 IMPLEMENTATION

### 3.1 CHUNKING ATTENTION AND MOE

Within the conventional Attention-MoE model structure (Figure 2), a bottleneck arises where the entire attention layer must be completed before the MoE layer can commence, rendering pipelining unfeasible. However, our key observation is that both the attention and MoE layers can be divided into chunks along the sequence dimension.

For attention, given $\mathbf{Q}, \mathbf{K}, \mathbf{V} \in \mathbb{R}^{N \times d}$, we can partition the computation as follows:

$$[\mathbf{O_1}, ..., \mathbf{O_n}]^T = \text{softmax}([\mathbf{Q_1}, ..., \mathbf{Q_n}]^T \mathbf{K}^T)\mathbf{V}$$

Where partitioning the computation does not affect the outcome:

$$\mathbf{O}_k = \text{softmax}(\mathbf{Q}_k \mathbf{K}^T)\mathbf{V}$$

Here, $\mathbf{O}_k$ and $\mathbf{Q}_k$ are both in $\mathbb{R}^{(N/n) \times d}$. Notably, this property is easy to notice and has been utilized by prior researchers to improve system throughput (Dao, 2023; Liu & Abbeel, 2023).

**Chunking Attention with Causal Mask:** Presently, most popular LLMs are structured as Decoder-only models. These models incorporate a causal mask, which is a lower triangular matrix, that should be applied to the softmax operation. FlashAttention (Dao et al., 2022; Dao, 2023) efficiently handles attention with a causal mask by disregarding zero entries. Nevertheless, merely partitioning and feeding the input sequence into it would yield incorrect results. Our approach involves implementing high-performance attention chunking for causal masks based on FlashAttention, correctly configuring the causal mask for each input chunk (Figure 5).

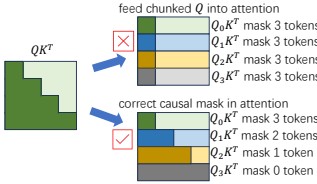

Figure 5: Chunking attention with causal mask: Attention shards need an offset on the causal mask.

For MoE, sharding the input batch along seqeunce dimension also preserves the token-wise computations of $p_i(x)$, $E_i(x)$ and thus $y$.

Consequently, we can chunk both attention and MoE as follows:

$$[\mathbf{O}_1, ..., \mathbf{O}_n]^T = \mathbf{MoE}(\mathbf{Attention}([\mathbf{Q}_1, ..., \mathbf{Q}_n]^T, \mathbf{K}, \mathbf{V}))$$

Each $\mathbf{O}_k$ depends solely on $\mathbf{Q}_k$, and the whole $\mathbf{K}$ and $\mathbf{V}$. $\mathbf{O}_k$ and $\mathbf{O}_j$ and their intermediate outcomes do not depend on each other for any $k \neq j$, allowing for pipelining of these chunks.

In our prior analysis, we overlooked layer normalization and other lightweight operations occurring between the Attention and MoE layers. These operations are also amenable to chunking, facilitating pipelining within a single transformer block. The pipeline step is shown below.

## 3.2  INTRA-BLOCK PIPELINING

To pipeline the attention and MoE layers, we leverage the capacity for concurrent computation and communication on NVIDIA GPUs. While previous implementations such as Tutel and FasterMoE focus on pipelining the MLP computation alongside MoE's all-to-all communication, we enhance speed by introducing pipelining of attention computation alongside MoE communication.

As illustrated in Figure 6, a better pipeline is achieved by overlapping all-to-all communication with both attention and MLP computation. For example, the attention computation towards getting $\mathbf{O_2}$ can be overlapped with the all-to-all in getting $\mathbf{O_1}$. This strategy maximizes throughput, mainly benefiting from the relatively longer execution times of these components.

In practice, the realized speedup may be moderated by the launch of multiple smaller kernels and unsatisfactory computation-communication overlapping. We will discuss the speedups (§4.1), the overheads (§4.2), and theoretical analysis (§C) further.

## 4  EVALUATION

We conducted a thorough assessment to evaluate the impact of deploying AMPIPE in the training of MoE-based LLMs. Through this evaluation, we aim to confirm that using AMPIPE significantly enhances the system's training throughput for MoE-based LLMs while preserving convergence.

**Testbed.** Our evaluations were conducted on two distinct testbeds. The first testbed is an AWS g5.24xlarge instance equipped with 4 NVIDIA A10G 24GB GPUs connected via PCIe. The second testbed comprised an A800 GPU server with 8 NVIDIA A800 SXM 80GB GPUs.

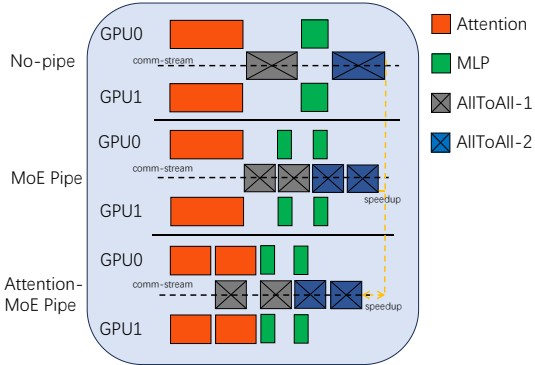

Figure 6: Intra-Block Pipeline Parallelism

**Setup.** As a baseline, we constructed MoE models based on GPT-3, leveraging the implementation in Megatron-LM (Shoeybi et al., 2019). We replaced the MLP layers in GPT-3 with MoE layers using the state-of-the-art Intra-MoE pipeline system Tutel (Hwang et al., 2023), following the Megatron-LM integration of Megablock (Gale et al., 2023). We instantiated these GPT-MoE models of varying sizes: base, medium, large, and xl (Table 1). This configuration effectively showcases the performance of AMPIPE for two key reasons: 1) it is firmly grounded in reality, as the majority of contemporary LLMs share a similar structure with GPT-3, and Megatron-LM is an industrial-level framework for LLM training, and 2) it provides sufficient hardware resources to highlight the capabilities of AMPIPE under different model configurations.

Table 1: The model settings of the evaluation.

| Model | head | embeding dim | mlp dim |
|---|---|---|---|
| GPT-MoE base | 12 | 768 | 3072 |
| GPT-MoE medium | 16 | 1024 | 4096 |
| GPT-MoE large | 20 | 1280 | 5120 |
| GPT-MoE xl | 25 | 1600 | 6400 |

### 4.1 PERFORMANCE OF AMPIPE WITH DATA PARALLELISM AND EXPERT PARALLELISM

In the previous section §2.3, we emphasized that AMPIPE's optimization is in transformer block and lies within the dimensions of data, expert, and tensor parallelism, orthogonal to optimizations related to pipeline parallelism. Now, our focus turns to the adaptation of AMPIPE within the commonly employed data and expert parallelism settings.

In this configuration, each GPU operates as an individual data and expert parallelism rank, typically used for moderate-scale training.

Figure 7 presents AMPIPE's performance analysis on the g5.24xlarge instance. The baseline configuration employs FlashAttention to achieve linear memory usage for extended sequences. We adjusted model size (base, medium, large and xl), sequence length (16K, 24K and 32K), batch size (1 and 2), and capacity factor (1, 2, and 4) for different benchmark cases after excluding those that couldn't fit on our testbed, resulting in a total of 56 cases. Notably, both AMPIPE and Tutel configurations also incorporate FlashAttention. Across these 56 benchmark cases, AMPIPE consistently surpasses Tutel, both employing a pipeline degree of four. When considering both forward and backward time in a transformer block, AMPIPE achieves an impressive 23% speedup on average compared to using FlashAttention alone, with peak speedups reaching 41%.

### 4.2 EFFECTIVENESS ANALYSIS OF AMPIPE

In this section, we first explore the impact of the pipeline degree on AMPIPE's performance in 56 benchmark cases. The pipeline degree is the only configuration parameter in the pipeline settings of Tutel and AMPIPE. It determines how many shards of computation and communication are partitioned into. A larger pipeline degree can potentially yield higher speedups as smaller shards can be overlapped more effectively. However, it also introduces greater overhead. Figure 8 illustrates how

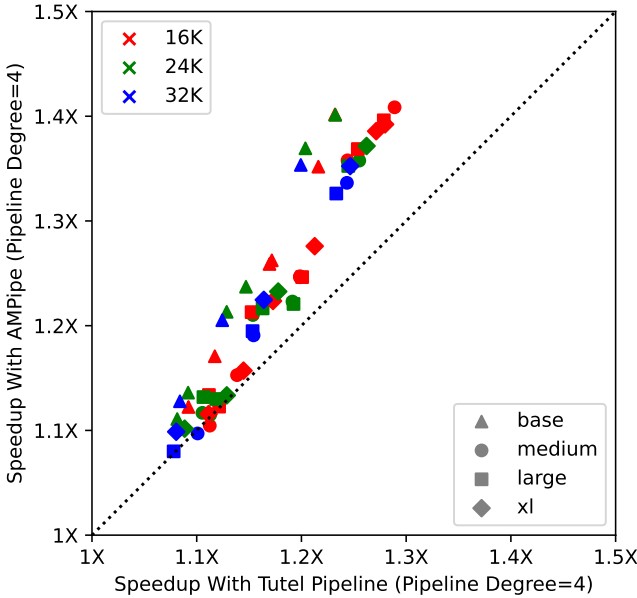

Figure 7: Speedup of both AMPIPE and Tutel w/ pipeline degree=4 across 56 benchmark cases when compared to the baseline settings of transformer blocks w/o pipeline, accounting for both forward and backward execution time of the transformer blocks.

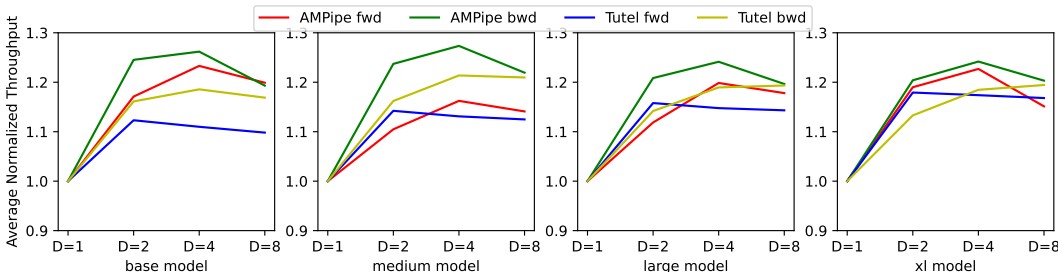

Figure 8: Speedup of AMPIPE and Tutel with different pipeline degrees

the pipeline degree influences speedups (labeled as "D" in the figure). Across these configurations, AMPIPE, with a pipeline degree of 4, consistently outperforms other settings. While Tutel's optimal performance varies with different pipeline degrees for different models, it still consistently lags behind AMPIPE's best performance.

We then investigate the overhead associated with AMPIPE's implementation. AMPIPE comprises two parts: chunking (as discussed in §3.1) and overlapping (as discussed in §3.2). Chunking introduces overhead primarily due to smaller but more GPU kernels (e.g., longer launch times). However, overlapping can mitigate these overheads, resulting in speedups. Figure 9 illustrates the slowdowns caused by chunking (i.e., w/o overlap, reflecting the overhead) and the speedups after overlapping (i.e., w/ overlap, reflecting the final speedups) across various settings. Our implementation incurs chunking overheads of less than 5% in all benchmark cases. Due to this minimal overhead, the speedup is largely realized, resulting in significant improvements over the original configuration. It's important to note that our implementation has not yet achieved theoretical speedups (§C).

## 4.3 END-TO-END PERFORMANCE OF AMPIPE

We evaluated AMPIPE's end-to-end performance on the A800 server. To emulate the real-world training task on a commodity cloud, we limit the data communication speed to 20GB/s by prolonging all-to-all communication. All models are configured with 12 layers of 8-degree data and expert parallelism to accommodate 80GB A800 GPUs. In this evaluation, AMPIPE showed an up to 16% speedup in throughput against the model setting without AMPIPE.

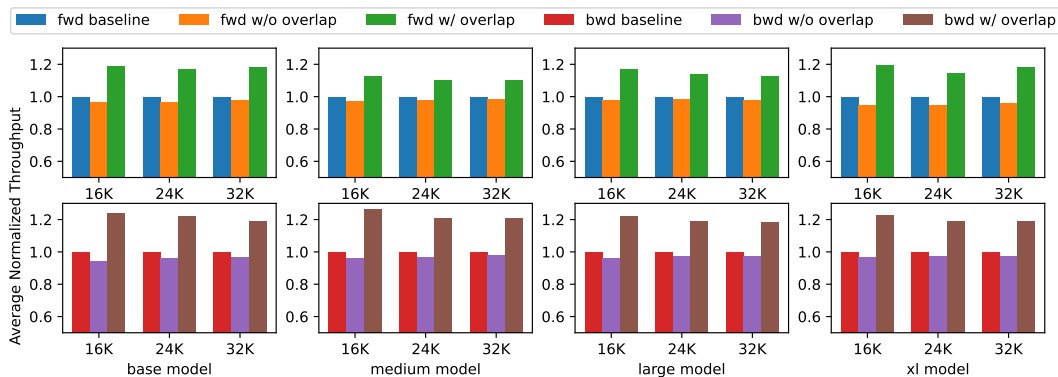

Figure 9: Overhead and final speedups of AMPIPE under different settings with a pipeline degree of 4 in benchmark cases.

Table 2: End-to-end training iteration time (ms) of GPT-MoE models on 8XA800 GPUs. AMPIPE reaches up to 16% speedup . We compare against a baseline running without AMPIPE. Both the baseline and AMPIPE activates FlashAttention.

| Model | MoE-medium 24K | MoE-medium 32K | MoE-large 24K | MoE-large 32K |
|---|---|---|---|---|
| Without AMPIPE | 2230.3 ms | 3207.7 ms | 2893.4 ms | 4082.4 ms |
| AMPIPE degree=2 | 2090.7 ms | 3048.5 ms | 2730.3 ms | 3972.8 ms |
| AMPIPE degree=4 | 1959.5 ms | 2767.8 ms | 2511.1 ms | 3650.2 ms |

## 4.4 AMPIPE'S IMPACT ON CONVERGENCE

In this section, we aim to highlight that AMPIPE does not affect the convergence of the original MoE model. We substantiate this claim by examining the training loss of AMPIPE, as presented in Figure 10.

The left figure in Figure 10 represents a configuration involving DP, EP, TP, and PP. In this setting, the model was trained on a Wikipedia dataset (Foundation) with a sequence length of 2k. On the other hand, the right sub-figure illustrates a scenario employing DP, EP, and PP. In this case, the model was trained using the CodeParrot dataset with a sequence length of 32k. Notably, the model trained with AMPIPE converges in an indistinguishable manner from the original setting without AMPIPE, affirming that AMPIPE's introduction maintains convergence fidelity.

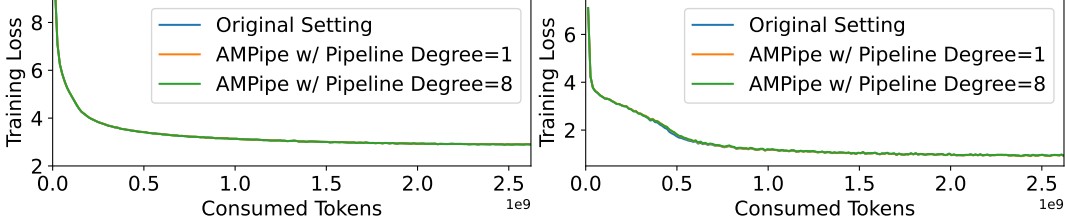

Figure 10: AMPIPE's Impact on Convergence

## 5 CONCLUSION

MoE has emerged as a pivotal technique facilitating large-scale sparse training for transformer models, garnering extensive attention within both the AI and system research communities. In this paper, we introduce AMPIPE, a novel and high-performance Intra-Block pipelining method. By strategically pipelining the attention layer, which incurs a quadratic time cost, alongside the sluggish all-to-all communication process, AMPIPE consistently surpasses existing pipeline and parallelism optimization techniques. Additionally, we conduct a meticulous analysis of speedup and overhead across various pipeline and parallelism configurations. Our results demonstrate that AMPIPE achieves an average speedup of 23% and can reach up to 41% speedups compared to basic transformer block setting without pipelining.

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

# A    LESSONS LEARNED

**Limitations** The implementation of AMPIPE is subject to two primary limitations. Firstly, following the chunking of the attention and MoE layers, additional overhead is introduced. Secondly, as AMPIPE is implemented in FP16 to enhance throughput, there is a possibility of increased accumulation after layer partitioning, potentially resulting in reduced numerical stability. However, it is noteworthy that the speedup achieved by AMPIPE effectively mitigates these overheads, as demonstrated in Figure 9. Furthermore, the observed numerical instability appears to have minimal impact, as illustrated in Figure 10.

**Future Applications** We anticipate that AMPIPE can significantly reduce the cost of training MoE-based LLM models within extended sequence contexts, achieving savings of up to 41%. Given the prevalent use of commercial cloud platforms for LLM training, AMPIPE has the potential to yield substantial time and cost savings. Moreover, we believe AMPIPE can serve as inspiration for the development of inference systems that offer lower latency and higher throughput.

# B    RELATED WORK

MoE models are getting more and more important in scaling dense LLMs. There are several feasible way to accelerate MoE-based LLMs.

**Optimizing Attention** Attention mechanisms are pivotal in Transformer models, but their quadratic time complexity poses a bottleneck in training long sequences. Previous efforts have focused on optimizing attention, such as transitioning to sparse attention paradigms (Child et al., 2019; Wang et al., 2020; Kitaev et al., 2020) or designing attention for long sequences (Zaheer et al., 2020; Guo et al., 2021; Ding et al., 2023; Beltagy et al., 2020). Recent work, such as FlashAttention (Dao et al., 2022; Dao, 2023), optimizes attention mechanisms without compromising convergence, achieving substantial speedups

**Optimizing MoE** Dedicated optimization of MoE layers is another effective method to boost MoE-based LLM performance. Approaches like FasterMoE (He et al., 2022), Tutel (Hwang et al., 2023), MPipeMoE (Zhang et al., 2023), and Lina (Li et al., 2023b) employ intra-MoE pipelines to overlap computation and communication costs. FasterMoE and FlexMoE (Nie et al., 2023) involve expert migration across GPUs to reduce all-to-all communication time. SmartMoE (Zhai et al., 2023) explores optimization opportunities within an expanded space of hybrid parallelism, while Kossmann et al. (2022) optimize MoEs using re-compilation techniques. Importantly, these designs are complementary to AMPIPE.

**Parallelism Trainings** Several parallelism strategies have been proposed to enhance distributed training efficiency. PyTorch-DDP (Li et al., 2020), ZeRO (Rajbhandari et al., 2020; Xu et al., 2020), and Petuum (Xing et al., 2015) target at efficient data parallelism. Gpipe (Huang et al., 2019), PipeDream (Narayanan et al., 2019), Terapipe (Li et al., 2021), vPipe (Zhao et al., 2021), and Naspipe (Zhao et al., 2022) optimize pipeline parallelism. Tofu (Wang et al., 2019), FlexFlow (Jia et al., 2019), and Varuna (Athlur et al., 2022) primarily focus on Tensor Parallelism. GShard (Lepikhin et al., 2020) introduces device-wise expert placement (expert parallelism) for MoE models, widely adopted in other MoE models and frameworks (Fedus et al., 2022; He et al., 2022; Hwang et al., 2023; Li et al., 2023b). Megatron-LM (Shoeybi et al., 2019; Narayanan et al., 2021; Korthikanti et al., 2023) implements high-performance 3D parallelism and enjoys widespread adoption in industry-level training. Alpa (Zheng et al., 2022) automates the discovery of parallelism solutions, incorporating both inter-operation and intra-operation parallelism, optimizing model data with an optimal parallelism strategy.

# C    SPEEDUP MODELING

This section constructs a model to estimate the theoretical speedup achieved by AMPIPE. Our model takes into consideration the attention layer, MLP, and all-to-all communication times. To simplify the analysis, we assume that all attentions are encoder attentions without masks, ensuring uniform execution times for all attention chunks.

We denote the execution time of the entire attention layer as $A$, with both the first and second all-to-all communication times set to $C$, and the execution time of the MLP as $M$. Additionally, we assume that the execution time of LayerNorm and other lightweight operators is represented as $O$.

In the absence of AMPIPE, the execution time of the entire transformer block is given by $A + 2C + M + O$.

Our overlapping strategy is implemented as outlined in §3.2. Within the MoE layer, two all-to-all communications are identified as a2a-1 and a2a-2. a2a-1 strives to overlap with the attention layer, except for the final shard, which overlaps with the MLP. Conversely, a2a-2 aims to overlap with the MLP layer. Consequently, when AMPIPE is configured with a pipeline degree of $p$, the execution time can be expressed as $A/p + \max(A/p, C/p) \times (p-1) + \max(M/p, C/p) \times p + C/p$. We assume that $O$ is fully overlapped. However, our evaluation reveals that the expected speedup consistently exceeds the observed speedup (Figure 11). This discrepancy can be attributed to various factors, including chunking overhead, incomplete overlap, host-bound kernel launches, and so on.

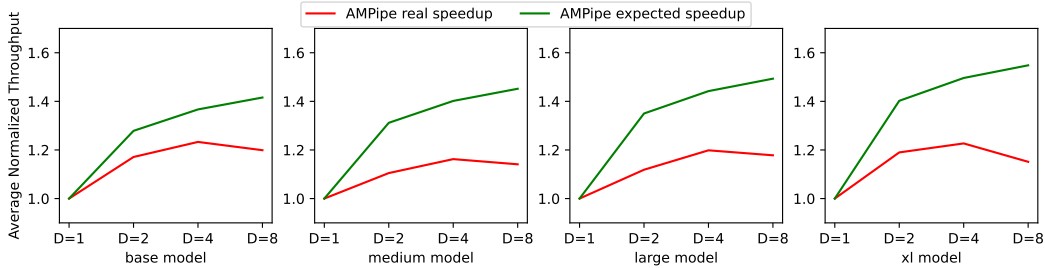

Figure 11: Expected and real speedup of AMPIPE under different settings.

