# OpenReview forum: "AMPipe: Accelerating MoE Model Training with Intra-Block Pipelining"
_ICLR.cc/2024/Conference — Submitted to ICLR 2024_

### Official Review · Reviewer_4gRQ · 2023-10-27

**Soundness:** 3 good
**Presentation:** 2 fair
**Contribution:** 2 fair
**Rating:** 5
**Confidence:** 4

**Summary:**

The authors propose AMPIPE, an optimization technique that improves the training speed of MoE models with pipeline parallelism. AMPIPE splits the attention computation and MLP computation into smaller chunks and schedules them to overlap with the all2all communication, reducing the cricial path in comparison to training an MoE model with pipeline parallelism enabled. The authors evaluate AMPIPE on several GPT-MoE models and show that it can achieve higher throughput than an existing MoE system Tutle without affecting convergence.

**Strengths:**

- The paper applies chunking to both Attention and MLP layers in a Transformer block to create a better pipeline scheme that exploits the overlapping potential between attention, MLP, and all2all communication in MoE models.
- The paper demonstrates the effectiveness of AMPIPE on low-performance interconnects.

**Weaknesses:**

1. The technical novelty of the work is limited. Particularly, the idea of breaking GeMM computation into smaller chunks and overlapping those chunks with communication collectives has been previously studied, such as in Tutle. The main contribution of the work is to extend that to the Attention layer in an MoE model.

2. There is a big misconnection between the motivation of the work and the evaluation. Pipeline parallelism for MoE models is usually used for massive MoEs (e.g. hundreds of billions or even trillion-scale models) when the model cannot fit on aggregated GPUs with data parallelism, expert parallelism, and tensor parallelism. However, the tested MoE models are relatively small, with the largest one having only 12 layers and a hidden dimension 1600. This raises questions in terms of whether pipeline parallelism is even needed for this scale of models, i.e., the authors might have compared with a very sub-optimal configurations that are unnecessarily slow or have compared with configs that do not need pipeline parallelism. Indeed, from Figure 8, we see that as the model size increases, the gap between AMPIPE and Tutle decreases. To be more convincing, the paper should either test larger-scale models where pipeline parallelism is actually needed or use stronger baselines that sweep over different parallelism combinations.

3. The paper lacks many important implementation details. In particular, while the projection of Q, K, V can be chunked along the sequence dimension, the softmax operation in the attention calculation requires all tokens in a sentence to calculate the normalized attention score. Therefore, it does not seem to be correct to claim that chunking Q does not affect the output of attention calculation unless additional synchronization is added. Also, the paper overlooks the normalization operations in its analysis, which are crucial for Transformer effectiveness and efficiency (as they may introduce additional synchronization that affects the pipeline's schedule. Unfortunately, the paper has been very vague on how these important operators are handled, e.g., in terms of how those operations are handled, the paper simply says "these operations are also amendable to chunking", but how and how would they affect the pipeline efficiency?

4. The experiment setup is sub-optimal. The model is evaluated on a single node with 4xA10G and 8xA800, but uses a pipeline depth of 4, which seems to be unnecessarily deep. The paper does not justify this choice of parallelism config and why this is a reasonable baseline. For the given model and hardware, is the parallelism config really a good choice in practice? It is not very difficult to improve a weak baseline, and not choose the best-performing baseline for the given model and hardware undermines the attractiveness of the proposed method.

**Questions:**

1. If DP + EP + TP already produces a configuration that provides much faster training speed why bother using pipeline parallelism?

2. Do the results in Figure 8 include normalization operations, such as softmax, layer norm, etc.

3. The paper simulates low data communication speed by adding artificial delays to all-to-all communication. How would the proposed technique perform on real commodity GPU clusters?

---

> ### Author Response · Authors · 2023-11-20
>
> Thank you very much for your insightful and constructive feedback, which has helped us improve our paper!
>
> **Weaknesses 1**: The technical novelty of the work is limited. Particularly, the idea of breaking GeMM computation into smaller chunks and overlapping those chunks with communication collectives has been previously studied, such as in Tutle. The main contribution of the work is to extend that to the Attention layer in an MoE model.
>
> **Response**: We appreciate your insights regarding the technical novelty of our work. Indeed, while the MoE communication and computation pipeline have been well explored in numerous previous studies (e.g., Tutel[1], FasterMoE[2], Lina[3], Mpipemoe[4]), our work uniquely tackles the MoE-attention bottleneck within the entire transformer block. This perspective, we argue, transcends mere incremental advancement, offering novel contributions to the field of Mlsys research.
>
> **Weaknesses 3 & Question 2**: The paper lacks many important implementation details...
>
> Do the results in Figure 8 include normalization operations, such as softmax, layer norm, etc.
>
> **Response**: Your question regarding the implementation details is important. Indeed, the results presented in Figure 8 encompass all elements within the transformer blocks, including both softmax and layernorm operations.
> **Normalization operations will not affect the integrity** of AMPipe. AMPipe's algorithm should be exactly the same as the original PyTorch implementation. We assure you of the correctness of our algorithm and will provide more comprehensive details in our revised manuscript. Regarding the concerns raised:
>
> (1) **Normalization Outside Softmax**: In modern LLM architectures like [GPT-2](https://github.com/huggingface/transformers/blob/main/src/transformers/models/gpt2/modeling_gpt2.py#L367C24-L367C33), [OPT](https://github.com/huggingface/transformers/blob/main/src/transformers/models/opt/modeling_opt.py#L260C40-L260C49), [LLaMA](https://github.com/huggingface/transformers/blob/main/src/transformers/models/llama/modeling_llama.py#L635C32-L635C44), and [Falcon](https://github.com/huggingface/transformers/blob/main/src/transformers/models/falcon/modeling_falcon.py#L732C28-L732C37), normalization operations (e.g., RMSNorm[5] and LayerNorm[6]) are typically applied **token-wise**. Consequently, when we chunk the sequence tokens into shards while retaining the integrity of each single token, the output remains unaffected.
>
> (2) **Normalization Inside Softmax**: These operations are conducted on a **query-wise** basis, meaning that each query token undergoes normalization across the entire kv length. Thus, dividing the whole query sequence into chunks also does not affect the output as we maintain the integrity of each query token's computation.
>
> Thus, both the normalization inside and outside softmax will not introduce additional synchronization that affects the pipeline's schedule.
>
> Additionally, the implementation of FlashAttention-v2[7], which includes two loops — the outer loop iterating over the query length and the inner loop handling softmax normalization across the kv length — demonstrates that chunking the outer loop does not impact the results. This method maintains the integrity of the output compared to the original attention implementation, providing further evidence for point (2).
>
> Furthermore, the Blockwise Parallel Transformer's[8] approach of splitting sequence along the query length also substantiates that such an operation does not compromise the model output, even with normalization operations. This approach offers validation for both points (1) and (2).

---

> > ### Author Response · Authors · 2023-11-20
> >
> > **Weaknesses 2 & Question 1**:
> > There is a big misconnection between the motivation of the work and the evaluation...
> >
> > If DP + EP + TP already produces a configuration that provides much faster training speed why bother using pipeline parallelism?
> >
> > **Response**: We would like to clarify that our experimental setup **does not employ a pipeline parallelism**. We categorize parallelism technologies into two categories: DP+EP+TP, which is used for optimizing single transformer blocks, and PP, which is used to schedule across multiple transformer blocks along the model depth. In this context, our proposed method, AMPipe, falls within the scope of DP+EP+TP, specifically targeting the optimization of a single transformer block.
> >
> > Due to resource constraints, our experiments were conducted without the implementation of pipeline parallelism. This allowed us to specifically assess the speed performance of our method in transformer block optimization. Regarding the **pipeline degree** mentioned in Figures 7 and 8, it refers to the **internal pipeline degree within AMPipe and Tutel** (i.e., intra-block pipeline as the paper title) in Figure 6 rather than an inter-block pipeline degree as in Figure 4. We intend to provide a more precise definition of 'pipeline degree' in our revised manuscript. For a pipeline degree of n and an original sequence length of N, each sequence chunk in our model is designed to have a length of N/n. This refined explanation will be included in the revised manuscript to enhance clarity and understanding.
> >
> >
> > **Weaknesses 4 & Question 3**:
> > The experiment setup is sub-optimal...
> >
> > The paper simulates low data communication speed by adding artificial delays to all-to-all communication. How would the proposed technique perform on real commodity GPU clusters?
> >
> >
> > **Response**: We appreciate your input on our emulation method. We emulated lower bandwidth by transmitting 20x the data volume. While Tutel[1] tested their speedup on high GPU-inter connections (up to 1600Gbps), it's common to train models on powerful GPUs like the A100 with lower inter-connection bandwidths (e.g., AWS's A100 server has 50GB/s[9] and Google Cloud's A100 server 12.5GB/s[10]). We admit the setting was sub-optimal for a very large-scale model. However, our aim was to simulate a realistic training environment under our limited budget, recognizing that it may not perfectly match all real-world deployments. Our experiments tested the case without a pipeline parallelism, as we believe that our algorithm is orthogonal to the pipeline parallelism. However, the trend observed should be reflective of most scenarios to some extent.
> >
> > [1] Changho Hwang, Wei Cui, Yifan Xiong, Ziyue Yang, Ze Liu, Han Hu, Zilong Wang, Rafael Salas,
> > Jithin Jose, Prabhat Ram, et al. Tutel: Adaptive mixture-of-experts at scale. Proceedings of
> > Machine Learning and Systems, 5, 2023.
> >
> > [2] Jiaao He, Jidong Zhai, Tiago Antunes, Haojie Wang, Fuwen Luo, Shangfeng Shi, and Qin Li. Fastermoe: modeling and optimizing training of large-scale dynamic pre-trained models. In Proceedings of the 27th ACM SIGPLAN Symposium on Principles and Practice of Parallel Programming,
> > pp. 120–134, 2022.
> >
> > [3] Jiamin Li, Yimin Jiang, Yibo Zhu, Cong Wang, and Hong Xu. Accelerating distributed {MoE}
> > training and inference with lina. In 2023 USENIX Annual Technical Conference (USENIX ATC
> > 23), pp. 945–959, 2023b.
> >
> > [4] Zheng Zhang, Donglin Yang, Yaqi Xia, Liang Ding, Dacheng Tao, Xiaobo Zhou, and Dazhao
> > Cheng. Mpipemoe: Memory efficient moe for pre-trained models with adaptive pipeline parallelism. In 2023 IEEE International Parallel and Distributed Processing Symposium (IPDPS),
> > pp. 167–177. IEEE, 2023.
> >
> > [5] ZHANG, Biao; SENNRICH, Rico. Root mean square layer normalization. Advances in Neural Information Processing Systems, 2019, 32.
> >
> > [6] BA, Jimmy Lei; KIROS, Jamie Ryan; HINTON, Geoffrey E. Layer normalization. arXiv preprint arXiv:1607.06450, 2016.
> >
> > [7] Tri Dao. Flashattention-2: Faster attention with better parallelism and work partitioning. arXiv
> > preprint arXiv:2307.08691, 2023.
> >
> > [8] LIU, Hao; ABBEEL, Pieter. Blockwise Parallel Transformers for Large Context Models. In: Thirty-seventh Conference on Neural Information Processing Systems. 2023.
> >
> > [9] https://aws.amazon.com/cn/ec2/instance-types/
> >
> > [10] https://cloud.google.com/blog/products/compute/a2-vms-with-nvidia-a100-gpus-are-ga

---

> ### Comment · Reviewer_4gRQ · 2023-12-04
> **Response to authors' rebuttal**
>
> Thank you for the detailed responses. While it is good that the authors clarified the pipeline degree definition in the paper, the reviewer is not fully convinced by the authors' other arguments.
>
> 1. The authors argue that normalization inside softmax does not require additional synchronization with a partitioning of the query sequence into chunks. However, this is not convincing. In models like GPT-2, the softmax in the attention mechanism does depend on the raw attention score of all tokens in a sentence, as shown here https://github.com/huggingface/transformers/blob/2c658b5a4282f2e824b4e23dc3bcda7ef27d5827/src/transformers/models/gpt2/modeling_gpt2.py#L208C10-L208C10. Therefore, the reviewer has a hard time to interpret the authors' response that this can be done without any synchronization. The authors may need to provide the details of how exactly the proposed scheme affects the softmax calculation in the attention at an operator level, e.g., how dividing the query sentence into chunks does not require synchronization while getting exactly the same results as the non-partitioned baseline.
>
> 2. The reviewer is not convinced that adding artificial network delay is the right methodology for conducting the experiments. If Google Cloud indeed has an environment that has low network bandwidth, why the results are not conducted on those platforms given that Google Cloud is publicly available?

---

### Official Review · Reviewer_wCG9 · 2023-10-29

**Soundness:** 2 fair
**Presentation:** 2 fair
**Contribution:** 2 fair
**Rating:** 5
**Confidence:** 4

**Summary:**

This work proposes AMPipe, a method to accelerate Mixture-of-Experts (MoEs) training when the sequence length is long. The basic idea is to split the sequence dimension into blocks so that the computation of one block can be overlapped with the communication of the next block.

**Strengths:**

- Training large language models for long sequence length is a timely and important topic.
- The design of computation-communication overlap is sound.

**Weaknesses:**

For me, this paper reads more of a technical report or technical blog introducing a nicely designed engineering effort. The major contribution is a chunking method along the sequence dimension to overlap the computation of FlashAttention and the All-to-All communication of MoEs for two consecutive chunks, which has limited novelty and takes barely one page (i.e., Section 3) to illustrate. I am afraid it may fail to provide significant insights to the community.

**Questions:**

(1) To support training on extremely long sequence length, memory is also important to consider. I would like to know whether AMPipe saves memory.

(2) Sequence parallel and tensor parallel are also important techniques to support lengthy sequences by amortizing the memory onto different GPUs. It would be interesting to see how to integrate AMPipe with them.

(3) Table 1 does not report the number of layers, the number of experts, and the size (number of parameters) of each model.

(4) According to the specification of A800, the NVLink bandwidth should be 400GB/s. It should be elaborated how the data communication speed is limited to 20GB/s in Section 4.3 (e.g., by transmitting 20x of data volume?) Moreover, although the intention to simulate commodity communication bandwidth is nice to have, I am afraid limiting the NVLink bandwidth to 1/20 of the origin (from 400GB/s to 20GB/s) is a good choice since the computing power (i.e., flops) of A800 is quite performant.

(5) If I understand well, AMPipe requires token-level communication in MoE layers, so I would like to ask whether it can be applied to sequence- or task-level MoEs?

---

> ### Author Response · Authors · 2023-11-20
>
> Thank you for your thoughtful feedback on our work. We appreciate the opportunity to clarify aspects of our research and methodology.
>
> **Question 1**: To support training on extremely long sequence length, memory is also important to consider. I would like to know whether AMPipe saves memory.
>
> **Response**: Your question regarding memory efficiency is pertinent. Building on the FlashAttention [1][2], AMPipe achieves linear memory usage with increasing sequence length. In our empirical evaluations, we observed no additional memory usage compared to current solutions, thus efficiently supporting training on extremely long sequences.
>
> **Question 2**: Sequence parallel and tensor parallel are also important techniques to support lengthy sequences by amortizing the memory onto different GPUs. It would be interesting to see how to integrate AMPipe with them.
>
> **Response**: AMPipe is theoretically compatible with data, tensor, and sequence parallelism[3]. We have already integrated it with tensor parallelism in our latest version, and details of this integration will be included in the appendix of a forthcoming revision. Regarding sequence parallelism, its integration is non-trivial. A more dedicated design is necessary to effectively combine current sequence parallelism computation and communication with the MoE computation and communication, aiming to achieve enhanced performance.
>
> **Question 3**: Table 1 does not report the number of layers, the number of experts, and the size (number of parameters) of each model.
>
> **Response**: Regarding the suggestion to add the number of layers and parameters in Table 1, we would like to clarify that our experiments in Sections 4.1 and 4.2 focus on speedup comparison within a transformer block instead of the whole model, emphasizing head and dimension settings. In this experiment, we use data and expert parallelism, and each GPU has one expert (varying expert numbers will only change local MLP, which is not a bottleneck in our evaluations). The base, medium, large, and xl transformer blocks have 7M, 13M, 20M, and 31M parameters, respectively. Additionally, we conducted an end-to-end experiment, as detailed in Section 4.3 and Table 2, using a cluster with eight A800 GPUs. We believe that our setting can reflect its real-world training potential to some extent.
>
> **Question 4**: According to the specification of A800, the NVLink bandwidth should be 400GB/s. It should be elaborated how the data communication speed is limited to 20GB/s in Section 4.3 (e.g., by transmitting 20x of data volume?) Moreover, although the intention to simulate commodity communication bandwidth is nice to have, I am afraid limiting the NVLink bandwidth to 1/20 of the origin (from 400GB/s to 20GB/s) is a good choice since the computing power (i.e., flops) of A800 is quite performant.
>
> **Response**: We appreciate your input on our bandwidth emulation method. We emulated lower bandwidth by transmitting 20x the data volume. While Tutel[4] tested their speedup on high GPU-inter connections (up to 1,600Gbps), it's common to train models on powerful GPUs like the A100 with lower network bandwidths (e.g., AWS's A100 server has 50GB/s[5] and Google Cloud's A100 server has 12.5GB/s[6]). Our aim was to simulate a realistic training environment under our limited budget, recognizing that it may not perfectly match all real-world deployments. However, the trend observed should be reflective of most scenarios.
>
> **Question 5**: If I understand well, AMPipe requires token-level communication in MoE layers, so I would like to ask whether it can be applied to sequence- or task-level MoEs?
>
> **Response**: AMPipe is adaptable to both sequence-level and task-level MoEs. Given that AMPipe effectively serves the finer-grained token-level communication, it can also be applied to these larger-grained scenarios. While it is particularly well-suited to token-level communication in MoEs, which involves shuffling tokens across different devices, it can also accommodate larger-grained communication in sequence or task-level MoEs, where tokens from the entire sequences or tasks are sent to the same experts/devices. Many LLM-based MoE models and systems (e.g., Tutel[4], FasterMoE[7], Lina[8]) utilize token-level communication, and our empirical experiments demonstrate AMPipe's effectiveness in this token-level scope.

---

> > ### Author Response · Authors · 2023-11-20
> >
> > [1] Tri Dao, Dan Fu, Stefano Ermon, Atri Rudra, and Christopher Re. Flashattention: Fast and memory- ´
> > efficient exact attention with io-awareness. Advances in Neural Information Processing Systems,
> > 35:16344–16359, 2022.
> >
> > [2] Tri Dao. Flashattention-2: Faster attention with better parallelism and work partitioning. arXiv
> > preprint arXiv:2307.08691, 2023.
> >
> > [3] Shenggui Li, Fuzhao Xue, Chaitanya Baranwal, Yongbin Li, and Yang You. 2023. Sequence Parallelism: Long Sequence Training from System Perspective. In Proceedings of the 61st Annual Meeting of the Association for Computational Linguistics (Volume 1: Long Papers), pages 2391–2404, Toronto, Canada. Association for Computational Linguistics.
> >
> > [4] Changho Hwang, Wei Cui, Yifan Xiong, Ziyue Yang, Ze Liu, Han Hu, Zilong Wang, Rafael Salas,
> > Jithin Jose, Prabhat Ram, et al. Tutel: Adaptive mixture-of-experts at scale. Proceedings of
> > Machine Learning and Systems, 5, 2023.
> >
> > [5] https://aws.amazon.com/cn/ec2/instance-types/
> >
> > [6] https://cloud.google.com/blog/products/compute/a2-vms-with-nvidia-a100-gpus-are-ga
> >
> > [7] Jiaao He, Jidong Zhai, Tiago Antunes, Haojie Wang, Fuwen Luo, Shangfeng Shi, and Qin Li. Fastermoe: modeling and optimizing training of large-scale dynamic pre-trained models. In Proceedings of the 27th ACM SIGPLAN Symposium on Principles and Practice of Parallel Programming,
> > pp. 120–134, 2022.
> >
> > [8] Jiamin Li, Yimin Jiang, Yibo Zhu, Cong Wang, and Hong Xu. Accelerating distributed {MoE}
> > training and inference with lina. In 2023 USENIX Annual Technical Conference (USENIX ATC
> > 23), pp. 945–959, 2023b.

---

### Official Review · Reviewer_aNzs · 2023-10-31

**Soundness:** 3 good
**Presentation:** 3 good
**Contribution:** 2 fair
**Rating:** 5
**Confidence:** 2

**Summary:**

Using the observation that the transformer attention layer can be chunked, AMPipe notes that the chunked computation can be pipelined with other work being done in a transformer network. Namely, the A2A data movement of MoE networks is can be dispached in chunks and overlapped with the chunked attention computation. This is similar to and builds on the Intra-MoE pipelining done by the tutel paper.

**Strengths:**

The work shows how pipelining the all2all MoE communication with attention computation can improve throughput by 1.4x over training with no pipelining (where the pipelining in tutel shows improvement of 1.3x).

**Weaknesses:**

- When taking into account the tutel baseline, AMPipe provides a 7% improvement at best? (1.4x vs 1.3x speedup). This is effectively low impact.
- The work does not show how this scales to large gpu clusters (where the majority of MoE training happens given MoE networks are relatively large).
- while novel, the novelty is limited to combining tutel based pipelineing with pipelining based on chunked attention computation (an already existing concept)
- This is generally more useful for long context length settings. But the majority of pretraining happens at seq len = 2k or 4k and long context length training only happens for a small subset of the total training time. This meta-point lowers the overall impact of works targeting improvements at really long context lengths.

**Questions:**

- Can a variant of figure 8 be creates which combines the fwd and bwd pass? this allows the reader to see the total end to end speedup at different degrees instead of just seeing it in table 2 in tabular form.

---

> ### Author Response · Authors · 2023-11-20
>
> Thank you very much for your insightful and constructive feedback, which has been instrumental in refining our paper.
>
> **Weakness 1**. When taking into account the tutel baseline, AMPipe provides a 7% improvement at best? (1.4x vs 1.3x speedup). This is effectively low impact.
>
> **Response**: Your observation about AMPipe's 7% improvement over the Tutel baseline is appreciated. However, it's important to note that the speedups in Figure 7 vary with different model/sequence settings. The maximum speedup compared to Tutel is indeed higher, as illustrated by the 1.4x vs 1.23x difference. Furthermore, in the context of training large LLMs on thousands of GPUs, which require months of training time and millions of dollars[1], even a 10% speedup can translate into substantial financial benefits, underscoring the practical significance of the AMPipe.
>
> **Weakness 2**. The work does not show how this scales to large gpu clusters (where the majority of MoE training happens given MoE networks are relatively large).
>
> **Response**: We appreciate your insight regarding scalability. We acknowledge that our evaluation, conducted on a smaller cluster, may not fully reflect the cases of training LLMs on real-world, large GPU clusters. We recognize that the communication bandwidth in larger clusters differs significantly from that in smaller setups. However, our present approach, encompassing diverse models, sequence settings, and low-bandwidth emulation, aims to replicate a range of real-world scenarios. We believe that performance in large GPU cluster environments will align with our experimental results to some extent, although perfect emulation may not be feasible.
>
> **Weakness 3**: while novel, the novelty is limited to combining tutel based pipelineing with pipelining based on chunked attention computation (an already existing concept)
>
> **Response**: Thank you for pointing out our work's conceptual scope. While there are numerous studies on MoE communication and computation pipeline with various methods(e.g., Tutel[2], FasterMoE[3], Lina[4], Mpipemoe[5]), our work distinctively addresses the MoE-attention bottleneck from a **whole transformer block perspective**. Our approach, we believe, is more than incremental work, offering valuable insights for future Mlsys research.
>
> **Weakness 4**: This is generally more useful for long context length settings. But the majority of pretraining happens at seq len = 2k or 4k and long context length training only happens for a small subset of the total training time. This meta-point lowers the overall impact of works targeting improvements at really long context lengths.
>
> **Response**: We acknowledge the prevalent trend of pretraining at shorter sequence lengths, a focus of prior MoE system works (e.g., Tutel, FasterMoE, Lina, Mpipemoe). However, AMPipe shows benefits in training with longer sequences, aligning with the emerging trend of longer sequences in LLM designs. We need to admit that addressing long context lengths is obtaining the importance, even though current methods often allocate a moderate portion of training time to them.
>
> **Question**: Can a variant of figure 8 be creates which combines the fwd and bwd pass? this allows the reader to see the total end to end speedup at different degrees instead of just seeing it in table 2 in tabular form.
>
> **Response**: We appreciate your suggestion to revise Figure 8 to combine forward and backward passes. We will incorporate this enhancement in our revised version. However, as discussed in the paper, Figure 8 is a variant of Figure 7 across 56 benchmarks. Pipeline degree is a configuration for AMPipe, and the case with the highest speedup is more important (i.e., pipeline degree=4 case in Figure 7).
>
>
> [1] https://blogs.nvidia.com/blog/what-are-large-language-models-used-for/
>
> [2] Changho Hwang, Wei Cui, Yifan Xiong, Ziyue Yang, Ze Liu, Han Hu, Zilong Wang, Rafael Salas,
> Jithin Jose, Prabhat Ram, et al. Tutel: Adaptive mixture-of-experts at scale. Proceedings of
> Machine Learning and Systems, 5, 2023.
>
> [3] Jiaao He, Jidong Zhai, Tiago Antunes, Haojie Wang, Fuwen Luo, Shangfeng Shi, and Qin Li. Fastermoe: modeling and optimizing training of large-scale dynamic pre-trained models. In Proceedings of the 27th ACM SIGPLAN Symposium on Principles and Practice of Parallel Programming,
> pp. 120–134, 2022.
>
> [4] Jiamin Li, Yimin Jiang, Yibo Zhu, Cong Wang, and Hong Xu. Accelerating distributed {MoE}
> training and inference with lina. In 2023 USENIX Annual Technical Conference (USENIX ATC
> 23), pp. 945–959, 2023b.
>
> [5] Zheng Zhang, Donglin Yang, Yaqi Xia, Liang Ding, Dacheng Tao, Xiaobo Zhou, and Dazhao
> Cheng. Mpipemoe: Memory efficient moe for pre-trained models with adaptive pipeline parallelism. In 2023 IEEE International Parallel and Distributed Processing Symposium (IPDPS),
> pp. 167–177. IEEE, 2023.

---

### Official Review · Reviewer_Ai23 · 2023-11-01

**Soundness:** 3 good
**Presentation:** 3 good
**Contribution:** 3 good
**Rating:** 6
**Confidence:** 2

**Summary:**

This paper introduces a method to speedup MoE inference and training by moving the pipeline parallelism to begin earlier in the layer's execution.  Specifically, the attention operation is split over queries along the sequence dimension into chunks which can be ran separately, before the same pipelining is applied to the feed forward networks in MoE layer.  This introduces a significant increase in throughput of 10-40% across many different configurations.  Code is provided where they add their AMPipe to the MegatronLM code.

**Strengths:**

* MoE is becoming very relevant for the current scaling of models as dense models are impractical at the trillion parameter scale, which makes this work relevant.
* Code is provided that is an extension of the MegatronLM code.  I skimmed their AMPipe implementation and it seems fairly simple which makes it easier to integrate into existing systems.
* They test on a variety of model scales and context lengths

**Weaknesses:**

* This work is conceptually very incremental, as they are simply moving the parallelism earlier in the layer's execution.  I am not familiar enough with the related work to evaluate how significant this work is in relation to others.

Nevertheless, I am voting to accept this paper as the quantitative speedups are significant and the method appears effective and simple enough to implement, which is very important in practice.

**Questions:**

**Setup** "As a baseline, we constructed MoE models based on GPT-3”
* GPT-3 is not an open source model, I believe you mean GPT-2, considering the description of the model in Table 1

**Table 1** I think it would be useful to the reader to add the # of layers and # of parameters to compare the different models

“With an implementation consisting of more than 1k lines of code (LoCs)”
* The fact that you provided code is great, but lines of code is not an informative metric.  The optimum number of lines of code is the minimum necessary to have performant and readable code.

It is worth explicitly defining what "pipe degree" is rather than keeping it implicit.  Perhaps say that pipe degree = n and chunk length = N / n, etc.

Typos:
“unofficially reported to use MoE paradigm.” - - - > “unofficially reported to use the MoE paradigm.”
For MoE, sharding the input batch along seqeunce dimension” - - > “For MoE, sharding the input batch along sequence dimension

---

> ### Author Response · Authors · 2023-11-20
>
> Thank you very much for your insightful and constructive feedback, which has helped us improve our paper.
>
> **Question**: Setup "As a baseline, we constructed MoE models based on GPT-3"
>
> **Response**: Thanks for pointing out our mistake. Our baseline is constructed on the open-sourced GPT-2, not GPT-3. This will be duly corrected in our revised manuscript.
>
> **Question**: Table 1 I think it would be useful to the reader to add the # of layers and # of parameters to compare the different models
>
> **Response**: Regarding the suggestion to add the number of layers and parameters in Table 1, we would like to clarify that our experiments in Sections 4.1 and 4.2 focus on speedup comparison within a transformer block instead of the whole model, emphasizing head and dimension settings. In this experiment, we use data and expert parallelism, and each GPU has one expert (varying expert numbers will only change local MLP time, which is not a bottleneck in our evaluations). The base, medium, large, and xl transformer blocks have 7M, 13M, 20M, and 31M parameters, respectively. Additionally, we conducted an end-to-end experiment, as detailed in Section 4.3 and Table 2, using a cluster with eight A800 GPUs.
>
> **Question**: "With an implementation consisting of more than 1k lines of code (LoCs)" The fact that you provided code is great, but lines of code is not an informative metric. The optimum number of lines of code is the minimum necessary to have performant and readable code.
>
> **Response**: Thanks for your constructive advice. We agree with your point on the metric of lines of code. Our revision will better reflect the emphasis on the performance and readability of the code rather than the quantity of code.
>
> **Question**: It is worth explicitly defining what "pipe degree" is rather than keeping it implicit. Perhaps say that pipe degree = n and chunk length = N / n, etc.
>
> **Response**: Thank you for highlighting the need to define "pipe degree" more explicitly. We confirm that your understanding is correct: with a pipe degree of n and an original sequence length of N, each sequence chunk will have a length of N/n. This definition will be included in the revised manuscript for clarity.
>
> **Question**: Typos: "unofficially reported to use MoE paradigm." - - - > "unofficially reported to use the MoE paradigm." For MoE, sharding the input batch along seqeunce dimension" - - > "For MoE, sharding the input batch along sequence dimension
>
> **Response**: Thanks for your diligence in identifying typographical errors. These will be corrected to ensure clarity and professionalism in our presentation.

---

> > ### Comment · Reviewer_Ai23 · 2023-11-21
> > **Response to authors 1**
> >
> > I thank the authors for their response.  I maintain my score and recommendation.

---

### Meta-Review · Area_Chair_khUh · 2023-12-15

**Metareview:**

The paper proposes AMPipe, a pipelining method to accelerate training of Mixture-of-Experts (MoE) models by overlapping the computation of attention layers with communication of MoE routers. The reviewers have raised several concerns:

(1) The novelty is a bit incremental over prior arts like Tutel, which has explored similar ideas for other layers. The key difference lies in handling attention layers by chunking attention computation along the sequence dimension. This constitutes the core technical contribution.

(2) The experiments use small models where pipeline parallelism may not even be needed. Whether the approach scales to massive models is unclear.

(3) Important details (e.g. handling of normalization layers) are missing, raising questions on correctness.

(4) The baseline is sub-optimal and the gap diminishes as model size increases. Using better baselines would have showcased the approach's advantages more clearly.

I would encourage the authors to consider a major revision, and the paper could be much stronger if these concerns could be properly addressed.

**Justification For Why Not Higher Score:**

NA

**Justification For Why Not Lower Score:**

NA

---

### Decision · Program_Chairs · 2024-01-16

Reject